# Leucocyte Telomere Length and Lung Cancer Risk: A Systematic Review and Meta-Analysis of Prospective Studies

**DOI:** 10.3390/cancers16183218

**Published:** 2024-09-21

**Authors:** Roberto Fabiani, Manuela Chiavarini, Patrizia Rosignoli, Irene Giacchetta

**Affiliations:** 1Department of Chemistry, Biology and Biotechnology, University of Perugia, 06123 Perugia, Italy; roberto.fabiani@unipg.it; 2Department of Biomedical Sciences and Public Health, Section of Hygiene, Preventive Medicine and Public Health, Polytechnic University of the Marche Region, 60126 Ancona, Italy; 3Local Health Unit of Bologna, Department of Hospital Network, Hospital Management of Maggiore and Bellaria, 40124 Bologna, Italy; irene.giacchetta@gmail.com

**Keywords:** lung cancer, leucocyte telomere length, prospective studies, meta-analysis

## Abstract

**Simple Summary:**

Telomere length (TL) may influence the carcinogenesis process. Short telomeres lead to genomic instability, which is an important event in tumor initiation, while long telomeres may promote cell division and immortality influencing tumor promotion/progression. Despite the numerous observational epidemiological studies available, the association between TL in leukocytes (LTL) and lung cancer risk is currently still uncertain. Therefore, we conducted this systematic review and meta-analysis of prospective studies to summarize the evidence and derive a more accurate estimate of the effect of LTL on lung cancer occurrence. Longer LTL could be a marker to identify subjects at high risk of developing lung cancer. This may help to focus secondary prevention (screening) on specific groups of subjects.

**Abstract:**

Although numerous epidemiological studies are available, the relationship between leukocyte telomere length (LTL) and lung cancer risk is still controversial. This systematic review and meta-analysis, performed according to the PRISMA statement and MOOSE guidelines, aims to summarize the evidence and calculate the risk of lung cancer associated with LTL. The literature search was performed on PubMed, Web of Science, and Scopus databases through May 2024. A random-effects model was used to calculate the pooled risk. Heterogeneity was assessed using I^2^ and Cochran’s Q statistic. Begg’s and Egger’s tests were used to detect publication bias. Based on 8055 lung cancer cases and 854,653 controls (nine prospective studies), longer LTL was associated with a significant 42% increment in all types of lung cancer risk (OR 1.42, 95% CI 1.24–1.63). The effect was even more evident for adenocarcinomas (OR 1.98, 95% CI 1.69–2.31), while no association was observed for squamous cell carcinoma (OR 0.87, 95% CI 0.72–1.06). Significantly, no association was found for current smokers (OR 1.08, 95% CI 0.90–1.30), while it remained high for both never-smokers (OR 1.92, 95% CI 1.62–2.28) and former smokers (OR 1.34, 95% CI 1.11–1.62). No significant publication bias was evidenced. Longer LTL is associated with an increment in lung cancer risk particularly in never-smoker subjects.

## 1. Introduction

Telomeres, small repetitive DNA sequences (5′-TTAGGG-3′) associated with a protein complex (shelterin proteins) at the end of chromosomes, have an important role for maintaining stability and integrity of genomes [1,2]. During cell division, because of the incomplete DNA duplication and in the absence of telomerases (an enzyme that catalyzes the addition of telomeric repeats), telomeres lose approximately 50–200 bp each mitotic cycle, becoming shorter [1,2]. When telomere length reaches a critical point, it signals the arrest of cell proliferation, the induction of cellular senescence, and apoptosis [3]. Telomere length (TL) is, thus, a retained “molecular clock”, useful as a biomarker of aging [4]. However, TL may be influenced by many other factors including gender [5], genetic polymorphisms [6], physical activity and smoking [7], air pollution and oxidative stress [8], nutritional status [9], inflammation [10], and cancer therapies [11]. Therefore, TL varies widely even among individuals with the same chronological age.

TL can be estimated either genetically, by identification of several TL-associated SNPs; or measured directly—generally in leukocytes (LTL) as easily obtainable surrogate cells—by various laboratory techniques [12]. Both genetically determined TL and measured LTL have been associated with different health outcomes including cardiovascular diseases, diabetes, Alzheimer’s disease, and various cancers [13,14]. However, the real association between LTL and cancer risk is controversial because telomeres may have dual roles in carcinogenesis: short telomeres lead to genomic instability, which is an important event in tumor initiation, while long telomeres may promote cell division and immortality, influencing tumor promotion/progression [15]. Indeed, previous meta-analyses have found no association between LTL and all cancers combined [16], breast [17], colorectal [18], prostate [19], and lung cancer risk [20]; however, others have found a positive association between short telomere length with an increased risk in all cancers combined [21], and gastrointestinal, head, and neck cancers [22]. Finally, longer LTL was associated with an increase in all cancer combined and lung cancer risks [23], as well as genetic variants related to longer telomeres associated with an increase in prostate cancer [24] and renal cell carcinoma [25].

Lung cancer is the most frequently diagnosed cancer and is still the leading cause of cancer death worldwide [26]. The temporal and spatial differences in the incidence and mortality of lung cancer reflect the different prevalence of smoking, which represents the main risk factor for this disease [26]. However, other important characteristics such as air pollution and an unhealthy diet can influence the risk of lung cancer with a significant impact, especially on non-smokers [27,28]. As reported above, in previous studies, the risk of lung cancer was not associated with LTL or was associated with a longer TL [20,23]. These differences may be due to several factors that may have an impact on TL measurement in different studies. In addition to the age of the subjects at the time of blood sampling and the accuracy of the methods used for TL assessment, the results may depend more profoundly on the different types of studies included. The true association between telomere length and cancer risk may be misled by the study design, whether case-control or cohort. It has been shown that both the presence of cancer and anticancer therapies can shorten telomere length [11]. Therefore, in case-control studies, which enroll subjects who have already developed the disease and who have likely been treated with anticancer therapies, the results may be biased and reflect reverse causality [23]. In other words, the shorter telomere in cases is due to the condition of this group of subjects that does not exist in controls. 

The present meta-analysis therefore aimed to investigate the association between LTL and lung cancer risk by selecting only those studies (prospective cohort studies and nested case-control studies) that used DNA samples collected before cancer diagnosis for LTL measurement. 

## 2. Materials and Methods

The present systematic review and meta-analysis was carried out and reported following the MOOSE (Meta-analysis Of Observational Studies in Epidemiology) guidelines and the PRISMA (preferred reporting items for systematic reviews and meta-analyses) statement [29,30]. The study protocol was registered in the International Prospective Register of Systematic Reviews (www.crd.york.ac.uk/PROSPERO/ (accessed on 1 March 2024), Registration No: CRD42024514266).

### 2.1. Systematic Search Strategy

A wide literature search was conducted up to May 2024 and without restrictions on the following databases: PubMed (http://www.ncbi.nlm.nih.gov/pubmed/ (accessed on 1 May 2024)), Web of Science (http://wokinfo.com/ (accessed on 1 May 2024)), and Scopus (https://www.scopus.com/ (accessed on 1 May 2024)). The PICO (Population, Intervention/exposure, Comparison, Outcome/event) framework was used to determine the eligibility of studies as follows: population (healthy adult participants or adult patients with lung cancer), exposure (measured telomere length in leucocyte DNA), comparison (longer leucocyte telomere length versus shorter leucocyte telomere length) and outcome (lung cancer, lung adenocarcinoma, lung squamous cell carcinoma). Articles of interest were identified using a combination of the following Medical Subject Headings (MeSH) terms and keywords: (cancer OR tumor OR tumour OR neoplasm OR “neoplastic disease”) AND “telomere length” AND lung. Furthermore, to find out additional publications of interest, we examined the reference list of included articles and recent relevant reviews.

### 2.2. Inclusion and Exclusion Criteria for Study Selection

Studies were considered eligible for inclusion if they met the following criteria: (i) investigated the relationship between measured LTL and lung cancer risk in adult subjects; (ii) presented a prospective study design (cohort studies and nested case-control studies) that used DNA samples extracted from peripheral blood and collected before cancer diagnosis; (iii) reported multivariate-adjusted lung cancer risk estimates (RR, OR, or HR) with 95% confidence intervals (CI) across LTL categories; (iv) the reference group in the categorical analysis being the longest or shortest LTL. 

Exclusion criteria were as follows: (i) LTL were genetically predicted by SNPs variants and/or Mendelian randomization; (ii) presented a case-control study design; (iii) reported data on lung cancer mortality; (iii) DNA was not extracted from blood; (iv) the subjects enrolled had another disease in addition to lung cancer. Duplicates, reviews, meta-analyses, comments, editorials, abstracts, summaries, and in vitro, animal, and ecological studies were excluded. Initially, the selection of articles was performed considering the title and abstract, and then the main text was evaluated for all studies not discarded in the first phase. Two authors (R.F. and M.C.) independently performed the study selection. Discrepancies were resolved by discussion with a third author (P.R.). The list of selected studies, the removal of duplicates, and the selection of studies of interest were managed with Zotero. Figure 1 shows the details of the study selection process.

### 2.3. Data Extraction and Quality Assessment

Two authors (R.F. and P.R.) independently extracted from each selected article the following information: first author, year of publication, location, study design and name, population characteristics (number of cases and controls, incident cases, length of follow-up, age), DNA source and extraction method, TL measurement method, histologic type of lung cancer, telomere length parametrization (units), OR/RR/HR (95% CI) according to gender and smoking habit, matched or adjusted variables, and quality of the study (NOS scores). When multiple estimates were reported in the article, those adjusted for the most confounding factors were extracted. The quality evaluation of the selected studies was performed according to the “Newcastle Ottawa Scale” (NOS) [31]. NOS used a star system, with a total score ranging from 0 to 9. An NOS score equal or superior to 7 indicated a high-quality study. Two investigators individually performed the quality evaluation of each selected study and disagreements were settled by a joint reevaluation of the original article with a third author.

### 2.4. Statistical Analysis

We estimated the association between LTL and lung cancer risk using the statistical program ProMeta version 3.0 (IDo Statistics-Internovi, Cesena, Italy). The relative risk and hazard ratio were taken as an approximation to the OR, and the meta-analysis was performed as if all types of ratios were ORs. The selected studies divided the LTL into tertile, quartile, and quintile. The combined risk estimates were calculated considering as reference the first quantile (shorter LTL) and the last quantile (longest LTL) using a random effect model. 

Heterogeneity between studies was evaluated by the chi-square-based Cochran’s Q statistic and the I^2^ statistic and considered significant if *p* < 0.05 or I^2^ > 50% [32,33]. 

Publication biases were detected by Begg’s and Egger’s tests [34,35]. Both methods were tested for funnel plot asymmetry—the former was based on the rank correlation between the effect estimates and their sampling variances, and the latter was based on a linear regression of a standard normal deviation on its precision. If a potential bias was detected, we further conducted a sensitivity analysis to assess the robustness of the combined effect estimates, the possible influence of the bias, and to have the bias corrected. We also conducted a sensitivity analysis to investigate the influence of a single study on the overall risk estimate by omitting one study in each turn. We considered the funnel plot to be asymmetrical if the intercept of Egger’s regression line deviated from zero with a *p*-value < 0.05.

## 3. Results 

### 3.1. Study Selection

As shown in Figure 1, a total of 1192 articles were identified from the initial search of three different databases.

After removing 445 duplicates, 747 articles remained for title and abstract analysis. Of these, 713 articles were excluded because they were inappropriate. Thirty-three records were selected and one was identified from the bibliography lists of already selected articles, so that thirty-four manuscripts were ultimately included for full-text analysis. Twenty-five items were excluded because they did not meet the inclusion criteria. Specifically, seven studies genetically predicted LTL using Mendelian randomization, five studies genetically predicted LTL using SNP variants, six studies used a retrospective case-control design, four studies did not report lung cancer risk, one study reported the risk of lung cancer mortality, one study did not extract DNA from blood, and one study looked at subjects with COPD (Chronic Obstructive Pulmonary Disease). The list of excluded articles with their reasons is given in Appendix A. Ultimately, nine articles [36,37,38,39,40,41,42,43,44] were selected for inclusion in the systematic review and meta-analysis (Figure 1).

### 3.2. Study Characteristics and Quality Assessment

The main characteristics of the included studies are reported in Table 1. Of the nine studies selected, four were conducted in China [37,39,40,43], two in the UK [36,38], two in the USA [41,42], and one in Finland [44]. Four were cohort studies [36,38,39,40] and five were nested case-control [37,41,42,43,44]. The DNA used to assess LTL was extracted from whole blood in eight studies, while in one study, DNA was extracted from buffy coats [43]. The DNA extraction methods used phenol–chloroform in three studies [37,43,44], cartridge-based magnetic bead in three studies [36,38,42], and QIAamp (Qiagen kit) in three studies [39,40,41]. Only one study used a Singleplex qPCR to determine LTL [41], while all the others used a Multiplex qPCR. In all studies, the LTL was expressed as a T/S ratio, which is the telomere repeat copy number (T) relative to that of a single copy gene (S). Risk estimates were calculated in relation to different LTL categories, tertiles were used in two studies [41,43], quartiles in four studies [36,37,42,44], and quintiles in three studies [38,39,40]. Seven studies reported the risk estimates for all types of lung cancer [36,37,38,41,42,43,44], all studies reported the risk estimated for adenocarcinoma, while the risk for squamous cell carcinoma was reported in six studies [36,38,40,41,42,44]. 

The results of the quality assessment, expressed as the NOS score, are reported in the last right column of Table 1. The quality score ranged from 6 to 9. Only one study reported a score of 6 [42], six studies reported a score of 7 [27,36,38,41,43,44], one study reported a score of 8 [39], and one study reported a score of 9 [40]. 

### 3.3. Meta-Analysis

Based on a total of 8055 lung cancer cases and 854,653 controls, the pooled estimates—comparing the longest LTL categories with the shortest—resulted in a statistically significant 42% increment in all types of lung cancer risk (OR 1.42, 95% CI 1.24–1.63) with moderate heterogeneity between studies (I^2^ =51%, *p* = 0.031) (Figure 2A). Even more evident was the increment in adenocarcinoma risk associated with the longest LTL, which was found to be 98% (OR 1.98, 95% CI 1.24–1.63) with no significant heterogeneity (I^2^ = 31%, *p* = 0.134) (Figure 2B). On the other hand, no significant association with squamous cell carcinoma was found (OR 0.87, 95% CI 0.72–1.06) (Figure 2C). 

The results of the stratified analysis according to gender, smoking habit, and the DNA extraction methods of lung cancer risk estimates associated with the longest LTL are shown in Table 2.

When all types of lung cancer were considered, the increment in risk in women was higher than in men (68% vs. 19%). Similar effects were observed for adenocarcinoma, the increment in risk was higher in women compared to men (114% vs. 75%). In the case of squamous cell carcinomas, no significant variations in risk were observed in both women and men (Table 2).

Regarding the smoking status, no association between longer LTL and all types of lung cancer was observed in current smokers, while the highest effect (92% increment in risk) was obtained in never-smokers (Table 2). For adenocarcinomas, a significant increment in risk was also noted in current smokers (78%), even if the highest effect was reported in never-smokers (121%). In regard to squamous cell carcinomas, no significant effect was observed in association with smoking status (Table 2).

The analysis of all types of lung cancer risk stratified by the DNA extraction methods shows that the phenol–chloroform method was associated with a higher estimate compared to both the magnetic beads and QIAamp methods. In this last case, the result was not statistically significant (Table 2). Instead, in the case of adenocarcinoma, the higher estimate was observed for the QUIamp method compared to both phenol–chloroform and magnetic beads methods (Table 2). Lung squamous cell carcinoma risk values did not display statistical significance for all three types of methods used to extract DNA (Table 2).

### 3.4. Sensitivity Analysis

Sensitivity analyses investigating the influence of a single study on the lung risk estimates suggested that these were not substantially modified by any single study. Indeed, the risk estimates for all types of lung cancer ranged from 1.32 (95% CI 1.18–1.46) when removing Wong et al. 2023 [36] to 1.48 (95% CI 1.24–1.63) when removing Han et al. 2023 [38]. The adenocarcinoma risk estimates ranged from 1.85 (95% CI 1.62–2.12) when removing Samavat et al. 2020 [39] to 2.05 (95% CI 1.72–2.44) when removing Han et al. 2023 [3]. Finally, the risk estimates ranged from 0.85 (95% CI 0.70–1.04) when removing Yuan et al. 2018 [40] to 0.92 (95% CI 0.74–1.13) when removing Han et al. 2023 [38] for squamous cell carcinoma.

### 3.5. Pubblication Bias

No significant publication bias was detected with both Egger’s and Begg’s methods, as shown by the *p* values (Table 2) and funnel plots (Appendix A).

## 4. Discussion

In the present meta-analysis, based on nine prospective studies, we found a strong association between longer LTL and overall lung cancer risk. The association was more pronounced for adenocarcinoma, in women, and in never-smoker subjects.

These findings are inconsistent with a previous meta-analysis by Karimi et al. [20], who found a non-statistically significant increase in overall lung cancer risk (OR = 1.13, 95% CI: 0.82–1.81) in association with shorter LTL. In addition, they found a smaller effect for adenocarcinoma (OR = 1.00, 95% CI: 0.68–1.47) and a bigger effect for squamous cell carcinoma (OR = 1.78, 95% CI: 1.25–2.53). These differences can essentially be explained by the different studies included in the two meta-analyses. The previous meta-analysis was based on a total of eight studies, including five case-control studies that were excluded from our meta-analysis. Moreover, we have included six additional prospective studies [36,37,38,39,40,41] that were published after the publication of Karimi’s article. We retain that the study design, prospective vs. retrospective, can significantly influence the results. In particular, the lag time of LTL measurement after the onset of lung cancer can increase the likelihood of reverse causation bias. For this reason, we excluded case-control studies in which LTL was measured in patients with the disease. Accordingly, our data are in agreement with another meta-analysis by Zhang et al. [23], who reported a statistically significant increase in lung cancer risk in association with longer LTL (OR: 1.690; 95% CI: 1.253–2.280), based on three prospective studies, which we also included in our meta-analysis [42,43,44].

Our results are further supported by other studies in which the LTL has been genetically determined. Indeed, several Mendelian randomization (MR) investigations—based on a random assortment of genetic variants—suggest a causal relationship between longer LTL and increased lung cancer risk in both Western [45,46] and Asian [47,48] populations. In particular, Zang et al. noted a highly statistically significant increment in lung adenocarcinoma risk per 1 kb increase in LTL (OR: 2.78; 95% CI 2.16–3.58) [45]. Similarly, Kachuri et al. by MR analysis estimated that a 1 kb increase in TL increases the risk of lung cancer (OR: 1.41; 95% CI 1.20–1.65) and lung adenocarcinoma (OR = 1.92; 95% CI 1.51–2.22) but not squamous lung carcinoma (OR: 1.04, 95% CI 0.83–1.29) [46]. Furthermore, similar results in different histological subtypes were reported by Cao et al., who showed that genetic estimation of a longer LTL was strongly associated with lung adenocarcinoma risk (OR: 2.69; 95% CI 2.11–3.42), while lung squamous cell carcinoma showed a marginal association (OR: 1.45; 95% CI 1.01–2.10) [48]. All these data are of particular importance because they should avoid reverse causality errors and not be altered by various confounding factors, which may not be fully controlled in observational studies. In addition, they are in agreement with our results, showing a strong effect on adenocarcinoma and marginal effect on squamous cell carcinoma.

Although in the meta-analysis we noted a low level of heterogeneity, a stratification analysis was carried out considering three important characteristics that may influence the association between LTL and lung cancer risk: gender, smoking habit, and the DNA extraction method. When stratified by gender, we found a stronger association of lung cancer risk in women compared to men. We have no explanation for this effect. We can consider that a meta-analysis showed that on average females had longer telomeres than males [5], and men may be more exposed to factors, including alcohol and smoking, that can shorten telomeres [49]. Indeed, we also found that the association among nonsmokers was higher than among smokers. It is possible that smoking—as a potent determinant of lung cancer—by causing the shortening of telomeres masks the effect of LTL on lung cancer risk estimates. In regard to the DNA extraction methods, we observed contrasting results for the QIAamp method, which produced a non-significant association for all types of lung cancer while being more effective toward adenocarcinoma. Indeed, different methods have been shown to produce pronounced differences in the relative telomere length [50]. In addition, together with DNA extraction methods, the experimental conditions (sample selection, collection, and storage) and, in particular, the assay procedures used to evaluate LTL are very important factors to consider [51]. However, in our meta-analysis, almost all of the studies selected (eight out of nine) used the same Multiplex qPCR method to measure telomere length. Therefore, we did not stratify according to this parameter. Further studies using standardized protocols for the assay itself are necessary to clarify all these methodological aspects.

Regarding the biological plausibility of our findings, it is not surprising that longer telomeres are associated with an increased risk of lung cancer. As mentioned above, the genetic determination of LTL is also consistent with our data. It should be considered that telomere length maintenance is a key step in tumorigenesis and a universal feature of immortalized tumor cells [52]. Cells possessing long telomeres may have a greater replicative capacity because signals produced by short telomeres are less effective in triggering senescence and apoptosis. This allows for continued cell division and increased DNA mutations that may be the basis for cancer development [53]. Some evidence suggests that long telomeres predispose to other cancers besides lung cancer, including acute myeloid leukemia, chronic lymphocytic leukemia, and melanoma [54]. Furthermore, longer leukocyte telomeres have been found to be a significant risk factor for the development of myeloproliferative neoplasms [54].

Our study presents several strengths and limitations. Unlike the previous meta-analysis, we selected only prospective studies, which are less subject to reverse causality bias than retrospective studies. Furthermore, the results were obtained from a substantial number of lung cancer cases, and no evident heterogeneity or significant publication biases were observed. All the included studies included very important characteristics, such as age, gender, and smoking habits in the matched or adjusted variables. This also enabled us to stratify the risk estimates according to gender and smoking habit.

However, due to several limitations, the results reported in this meta-analysis should be interpreted with caution. First, the included studies were observational and, although the quality was quite high (as evidenced by the NOS analysis), they may be subject to risk of bias due to misclassification of the included subjects. Second, although the method used to measure LTL was the same for most studies (Multiplex qPCR), the definition of relative telomere length (reported as T/S ratio) varied among the included studies. This may compromise the reliability of the results and prevent definitive conclusions. Third, the categorization of LTLs into different quantiles (tertiles, quartiles, and quintiles) in various studies can be a further source of data variability and uncertainty. Finally, the use of leukocytes to determine TL may not be truly representative of TL in target lung cells. TL varies between organs and tissue types within an individual, although it has been observed to be highly correlated [55]. However, it is evident that no study has assessed telomere length in cells that are difficult to obtain via invasive procedures.

## 5. Conclusions

Considering only prospective studies, a longer LTL was significantly associated with an increased risk of lung cancer. The effect was more evident for the adenocarcinoma subtype, in women, and in never-smokers. Due to the complex role that telomere length may play in this phenomenon, further larger studies—adequately designed and standardized for different analytical methods—are needed to validate these results and highlight the biological mechanisms that may be involved.

## Figures and Tables

**Figure 1 cancers-16-03218-f001:**
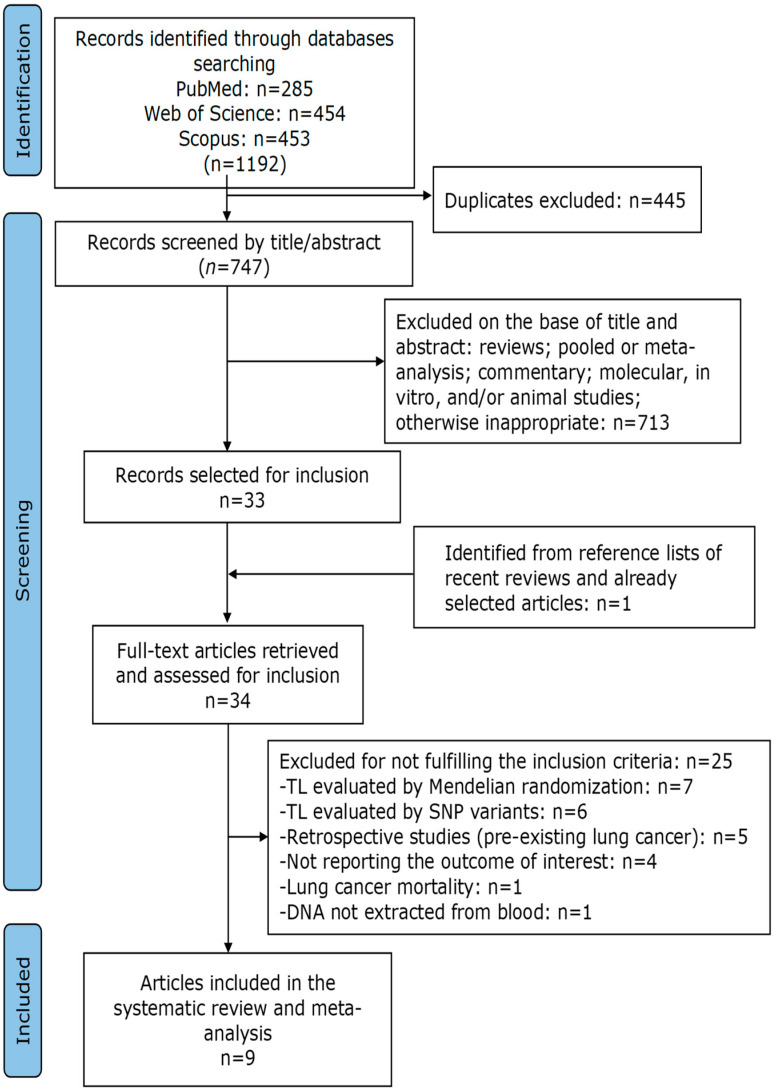
PRISMA flowchart of the selection process of included studies.

**Figure 2 cancers-16-03218-f002:**
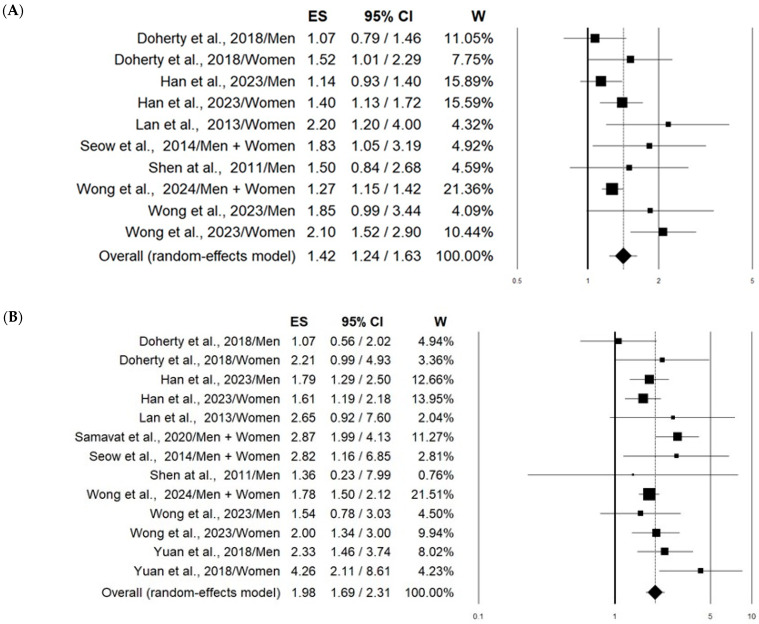
Forest plots showing the association between telomere length (comparing longest vs. shortest) in peripheral blood leukocytes and risk of all types of lung cancer (**A**) [36,37,38,41,42,43,44], adenocarcinoma (**B**) [36,37,38,39,40,41,42,43,44], and squamous cell carcinoma (**C**) [36,38,40,41,42,44].

**Table 1 cancers-16-03218-t001:** Characteristics of the studies included in the meta-analysis on the association between LTL and lung cancer risk.

First Author Year Location Reference	Study Design and Name Population Cases/Controls Incident Cases Follow-Up Age	DNA Source and Extraction Method	TL ^1^ Measurement Method and Units to Express TL	Histologic Type of Lung Cancer	Telomere Length Parametrization (Units)	OR/RR/HR (95% CI) According to Gender and (Smoking)	Matched or Adjusted Variables	NOS
Wong et al. 2024 UK [36]	Cohort UK Biobank 371,890 subjects Incident cases: 2829 Follow-up: 12.36 ± 1.64 years Age: 40–69 years	Peripheral blood Cartridge-based magnetic bead	Multiplex qPCR T/S ratio ^2^	All types Adenocarcinoma Squamous cell carcinoma	Quartile 1 (T/S < 0.74) Quartile 4 (T/S ≥ 0.91) Quartile 4 Quartile 4 Quartile 4 Quartile 4 Quartile 4 Quartile 4 Quartile 4 Quartile 4 Quartile 4 Quartile 1 Quartile 4 Quartile 1 Quartile 4	MEN + WOMEN (All) 1.00 (Ref.) 1.27 (1.15–1.42) MEN + WOMEN (Never-smokers) 1.72 (1.31–2.26) MEN + WOMEN (Former smokers) 1.33 (1.13–1.56) MEN + WOMEN (Current smokers) 1.09 (0.92–1.30) MEN (Never-smokers) 2.03 (1.27–3.26) MEN (Former smokers) 1.19 (0.96–1.48) MEN (Current smokers) 1.19 (0.94–1.51) WOMEN (Never-smokers) 1.55 (1.11–2.17) WOMEN (Former smokers) 1.57 (1.23–2.02) WOMEN (Current smokers) 0.92 (0.71–1.20) MEN + WOMEN (All) 1.00 (Ref.) 1.78 (1.50–2.12) MEN + WOMEN (All) 1.00 (Ref.) 0.88 (0.67–1.17)	Age, sex, race/ethnicity, smoking, assessment center, body mass index, Townsend deprivation index, alcohol, secondhand smoke, leucocyte differentials	7
Wong et al. 2023 China [37]	Nested case-control on -Shanghai Women’s Health Study (SWHS) Cases: 798 Age: 56.8 ± 8.9 years Controls: 792 Age: 56.9 ± 8.9 years -Shanghai Men’s Health Study (SMHS) cohort Cases: 161 Age: 62.3 ± 8.4 years Controls: 162 Age: 62.7 ± 8.5 years	Whole blood Phenol–chloroform	Multiplex qPCR T/S ratio	All types Adenocarcinoma	Quartile 1 (T/S < 0.487) Quartile 4 (T/S > 0.724) Quartile 1 (T/S < 0.487) Quartile 4 (T/S > 0.724) Quartile 1 Quartile 4 Quartile 1 Quartile 4	WOMEN (Never-smokers) 1.00 (Ref.) 2.10 (1.52–2.90) MEN (Never-smokers) 1.00 (Ref.) 1.85 (0.99–3.44) WOMEN (Never-smokers) 1.00 (Ref.) 2.00 (1.34–3.00) MEN (Never-smokers) 1.00 (Ref.) 1.54 (0.78–3.03)	Age, BMI, education, alcohol, secondhand smoke	7
Han et al. 2023 UK [38]	Cohort UK Biobank 425,146 subjects 226,072 Women 199,074 Men Incident cases: 1909 Follow-up: median 13 years Age: 37–73 years	Peripheral blood Cartridge-based magnetic bead	Multiplex qPCR T/S ratio	All types Adenocarcinoma Squamous cell carcinoma Others	Quintile 1 (Short TL) Quintile 5 (Long TL) Quintile 5 Quintile 5 Quintile 5 Quintile 5 Quintile 1 Quintile 5 Quintile 5 Quintile 5 Quintile 5 Quintile 5 Quintile 1 Quintile 5 Quintile 5 Quintile 5 Quintile 5 Quintile 5 Quintile 1 Quintile 5	MEN + WOMEN (All) 1.00 (Ref.) 1.25 (1.09–1.45) WOMEN (All) 1.40 (1.13–1.72) MEN (All) 1.14 (0.93–1.40) MEN + WOMEN (Never-smokers) 2.14 (1.46–3.15) MEN + WOMEN (Previous smokers) 1.15 (0.98–1.34) MEN + WOMEN (All) 1.00 (Ref.) 1.69 (1.35–2.11) WOMEN (All) 1.61 (1.19–2.18) MEN (All) 1.79 (1.29–2.50) MEN + WOMEN (Never-smokers) 2.31 (1.38–3.86) MEN + WOMEN (Previous smokers) 1.55 (1.20–1.99) MEN + WOMEN (All) 1.00 (Ref.) 0.75 (0.52–1.08) WOMEN (All) 0.84 (0.48–1.49) MEN (All) 0.69 (0.43–1.10) MEN + WOMEN (Never-smokers) 3.83 (0.73–19.99) MEN + WOMEN (Previous smokers) 0.69 (0.47–1.00) MEN + WOMEN (All) 1.00 (Ref.) 1.16 (0.92–1.45)	Age, sex, smoking, ethnicity, BMI, alcohol status, white blood cell count, educational level, household income, Townsend deprivation index	7
Samavat et al. 2020 China [39]	Cohort Singapore Chinese Health Study (SCHS) 28,125 subjects Incident cases: 309 Age: 62.9 ± 7.7 years Follow-up: 13.25 years	Whole blood QIAamp ^3^	Multiplex qPCR T/S ratio	Adenocarcinoma	Quintile 1 (T/S 0.7–0.8) Quintile 5 (T/S 1.24–1.42)	MEN + WOMEN (All) 1.00 (Ref.) 2.87 (1.99–4.13)	Age, sex, dialect group, BMI, level of education, smoking status, number of cigarettes per day, number of years of smoking, alcohol consumption, weekly vigorous work or strenuous sports, number of hours of sleep per night, diabetes mellitus, hypertension	8
Yuan et al. 2018 China [40]	Cohort Singapore Chinese Health Study (SCHS) 26,540 subjects Incident cases: 654 Age: 45–74 years Follow-up: 11,8 years	Peripheral blood QIAamp	Multiplex qPCR T/S ratio	Adenocarcinoma Squamous cell carcinoma Others	Quintile 1 (T/S 0.19–0.83) Quintile 5 (T/S 1.19–3.24) Quintile 5 Quintile 5 Quintile 5 Quintile 5 Quintile 1 Quintile 5 Quintile 1 Quintile 5	MEN + WOMEN (All) 1.00 (Ref.) 2.84 (1.94, 4.15) MEN + WOMEN (Never-smokers) 3.14 (1.80, 5.49) MEN + WOMEN (Ever smokers) 2.46 (1.45, 4.18) MEN (All) 2.33 (1.46, 3.74) WOMEN (All) 4.26 (2.11, 8.61) MEN + WOMEN (All) 1.00 (Ref.) 1.13 (0.60, 2.13) MEN + WOMEN (All) 1.00 (Ref.) 1.05 (0.69, 1.58)	Age, sex, dialect group, education, body mass index, number of cigarettes per day, number of years of smoking, number of years since quitting smoking (for former smokers only), and alcohol consumption	9
Doherty et al. 2018 USA [41]	Nested case-control on β-Carotene and Retinol Efficacy Trial (CARET) cohort Cases: 709 Age: 59.3 ± 5.5 years Controls: 1313 Age: 60.3 ± 5.4 years	Blood QIAamp	Singleplex qPCR T/S ratio	All types Adenocarcinoma Squamous cell carcinoma Small cell carcinomas	Tertile 1 (shortest TL) Tertile 3 (longest TL) Tertile 3 Tertile 3 Tertile 3 Tertile 3 Tertile 1 Tertile 3 Tertile 3 Tertile 3 Tertile 3 Tertile 3 Tertile 1 Tertile 3 Tertile 3 Tertile 3 Tertile 3 Tertile 3 Tertile 1 Tertile 3 Tertile 3 Tertile 3 Tertile 3 Tertile 3	MEN + WOMEN (All) 1.00 (Ref.) 1.21 (0.95–1.55) MEN + WOMEN (Former smokers) 1.82 (1.20–2.77) MEN + WOMEN (Current smokers) 0.96 (0.71–1.31) WOMEN (All) 1.52 (1.01–2.29) MEN (All) 1.07 (0.79–1.46) MEN + WOMEN (All) 1.00 (Ref.) 1.45 (0.88–2.37) MEN + WOMEN (Former smokers) 2.26 (1.03–4.96) MEN + WOMEN (Current smokers) 1.10 (0.58–2.12) WOMEN (All) 2.21 (0.99–4.93) MEN (All) 1.07 (0.56–2.02) MEN + WOMEN (All) 1.00 (Ref.) 0.96 (0.54–1.70) MEN + WOMEN (Former smokers) 1.82 (0.50–6.58) MEN + WOMEN (Current smokers) 0.86 (0.44–1.69) WOMEN (All) 1.51 (0.53–4.29) MEN (All) 0.81 (0.40–1.65) MEN + WOMEN (All) 1.00 (Ref.) 0.92 (0.51–1.66) MEN + WOMEN (Former smokers) 0.92 (0.34–2.52) MEN + WOMEN (Current smokers) 0.86 (0.41–1.83) WOMEN (All) 0.83 (0.33–2.11) MEN (All) 0.91 (0.41–2.01)	Age, smoking status, sex, race/ethnicity, enrollment year, asbestos exposure, and follow-up time	7
Seow et al. 2014 USA [42]	Nested case-control on Prostate, Lung, Colon, and Ovarian (PLCO) cohort Cases: 403 Age: 64.07 ± 4.97 years Controls: 403 Age: 63.64 ± 4.74 years Only data from PLCO were included to avoid repeated reports	Whole blood PLCO: magnetic bead-based extraction	Multiplex qPCR T/S ratio	All types Adenocarcinoma Squamous cell carcinoma	Quartile 1 (T/S <0.99) Quartile 4 (T/S >1.30) Quartile 4 Quartile 4	MEN + WOMEN (All) 1.00 (Ref.) 1.83 (1.05–3.19) MEN + WOMEN (All) 2.82 (1.16–6.85) MEN + WOMEN (All) 1.57 (0.44–5.56)	Age, pack-years of smoking	6
Lan et al. 2013 China [43]	Nested case-control on Shanghai Women’s Health Study cohort (SWHS). Cases: 215 Controls: 215 Age: 40–70 years	Buffy coats Phenol–chloroform	Multiplex qPCR T/S ratio	All types Adenocarcinoma ^4^	Tertile 1 (T/S <1.37) Tertile 3 (T/S ≥1.60) Quartile 1 (T/S <1.30) Quartile 4 (T/S >1.64)	WOMEN (All) 1.00 (Ref.) 2.20 (1.20–4.00) WOMEN (All) 1.00 (Ref.) 2.65 (0.92–7.60)	Age, ever smoking	7
Shen et al. 2011 Finland [44]	Nested case-control on Alpha-Tocopherol, Beta-Carotene Cancer Prevention (ATBC) Cases: 229 Age: 59 ± 5 years Controls 229 Age: 58 ± 5 years	Whole blood Phenol–chloroform	Multiplex qPCR T/S ratio TL cases: 1.14 ± 0.23 TL controls: 1.10 ± 0.22	All types Adenocarcinoma ^4^ Squamous cell carcinoma ^4^	Quartile 1 (T/S ≤0.94) Quartile 4 (T/S >1.25) Quartile 4 Quartile 4	MEN (Smokers) 1.00 (Ref.) 1.50 (0.84–2.68) MEN (Smokers) 1.36 (0.23–7.99) MEN (Smokers) 0.66 (0.21–2.07)	Age, number of cigarettes per day, and number of years smoked	7

^1^ Telomere length; ^2^ ratio of the telomere repeat copy number (T) relative to that of a single copy gene (S); ^3^ by Qiagen; ^4^ data reported in Ref. [42].

**Table 2 cancers-16-03218-t002:** Results of stratified analysis of lung cancer risk estimates associated with the longest LTL.

	No. of Studies	References	No. of Estimates	Combined Risk Estimate		Test of Heterogeneity			Publication Bias	
				Value (95% CI)	*p*	Q	I^2^%	*p*	Egger *p*	Begg *p*
**All types of lung cancer**										
All	7	[36,37,38,41,42,43,44]	10	1.42 (1.24–1.63)	<0.0001	18.36	50.98	0.031	0.072	0.089
Men	4	[37,38,41,44]	4	1.19 (1.00–1.40)	0.046	3.16	5.05	0.368	0.157	0.174
Women	4	[37,38,41,43]	4	1.68 (1.33–2.13)	<0.0001	5.41	44.53	0.144	0.332	0.497
Current smokers	3	[36,41,42]	4	1.08 (0.90–1.30)	0.404	4.69	35.99	0.196	0.570	0.497
Never-smokers	4	[36,37,38,42]	6	1.92 (1.62–2.28)	<0.0001	2.33	0.00	0.801	0.682	0.573
Former smokers	3	[36,38,41]	4	1.34 (1.11–1.62)	0.003	7.54	60.23	0.056	0.121	0.174
DNA extraction method										
QUIamp	3	[39,40,41]	2	1.24 (0.88–1.74)	0.213	1.81	44.69	0.179	-	-
Magnetic beads	3	[36,38,42]	4	1.28 (1.16–1.42)	<0.0001	3.54	15.26	0.316	0.471	0.174
Phenol–chloroform	3	[37,43,44]	4	1.96 (1.55–2.49)	<0.0001	1.17	0.00	0.761	0.491	0.497
**Adenocarcinoma**										
All	9	[36,37,38,39,40,41,42,43,44]	13	1.98 (1.69–2.31)	<0.0001	17.42	31.13	0.134	0.361	0.272
Men	5	[37,38,40,41,44]	5	1.75 (1.38–2.20)	<0.0001	3.91	0.0	0.418	0.485	0.624
Women	5	[37,38,40,41,43]	5	2.14 (1.55–2.96)	<0.0001	6.71	40.43	0.152	0.146	0.142
Current smokers	3	[40,41,42]	3	1.78 (1.05–3.01)	0.033	3.61	44.63	0.164	0.833	0.602
Never-smokers	4	[37,38,40,42]	5	2.21 (1.72–2.38)	<0.0001	2.90	0.0	0.574	0.930	1.000
Former smokers	2	[38,41]	2	1.61 (1.26–2.04)	0.0001	0.80	0.0	0.371	-	-
DNA extraction method										
QUIamp	3	[39,40,41]	5	2.35 (1.59–3.47)	<0.0001	9.61	58.38	0.047	0.680	0.624
Magnetic beads	3	[36,38,42]	4	1.77 (1.54–2.02)	<0.0001	1.44	0.00	0.696	0.425	0.497
Phenol–chloroform	3	[37,43,44]	4	1.91 (1.38–2.64)	<0.0001	0.95	0.00	0.814	0.816	1.000
**Squamous cell carcinoma**										
All	6	[36,38,40,41,42,44]	8	0.87 (0.72–1.06)	0.171	3.77	0.0	0.806	0.375	0.458
Men	3	[38,41,44]	3	0.72 (0.50–1.04)	0.079	0.16	0.0	0.923	0.920	0.602
Women	2	[38,41]	2	0.96 (0.58–1.58)	0.871	0.93	0.0	0.334	-	-
Current smokers	2	[41,42]	2	0.82 (0.48–1.38)	0.450	0.06	0.0	0.805	-	-
Never-smokers	1	[38]	1	3.83 (0.73–19.99)	-	-	-	-	-	-
Former smokers	2	[38,41]	2	0.91 (0.39–2.17)	0.838	2.00	50.11	0.157	-	-
DNA extraction method										
QUIamp	3	[39,40,41]	3	1.05 (0.68–1.61)	0.825	1.03	0.00	0.597	0.622	0.602
Magnetic beads	3	[36,38,42]	4	0.84 (0.68–1.05)	0.126	1.71	0.00	0.634	0.651	0.497
Phenol–chloroform	3	[37,43,44]	1	0.66 (0.21–2.07)	-	-	-	-	-	-

## Data Availability

The data presented in this study are available in this article.

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
