# Peer review of "Leucocyte Telomere Length and Lung Cancer Risk: A Systematic Review and Meta-Analysis of Prospective Studies"

_cancers, 2024, doi:10.3390/cancers16183218_

Round 1

Reviewer 1 Report

Comments and Suggestions for Authors

Fabiani et al. have performed a systematic review and meta-analysis of prospective studies to examine the association between leucocytes telomere length (LTL) and the risk of developing lung cancer. According to their study, longer LTL is resulted associated to an increment of lung cancer risk, particularly in never smoker subjects.

Overall, the manuscript is very well written, methodology is clearly described with particular regard to sensitivity analyses and detailed quality assessment of the included and excluded papers. Additionally, limitations are properly highlighted in the manuscript.

The major criticism is that there is neither completely novel to the cancer field nor the authors explained how telomere length of leucocytes could be considered a good risk factor for preventive interventions in lung cancer occurrence as anticipated by the authors in the summary section.

However, considering the rigorous approach of the analysis performed by the authors, the present study merits publication after including additional comments on the rationale and implication of leucocytes telomere length as risk factor for lung cancer incidence.

Author Response

We agree with the reviewer that the last sentence of the summary section is unclear and may lead to misunderstandings. In fact, we did not state that "telomere length of leucocytes could be considered a good risk factor for...", we meant that "longer LTL could be a marker to identify subjects at high risk of developing lung cancer". Accordingly, we have corrected the sentence (highlighted in red) in the revised manuscript.

Reviewer 2 Report

Comments and Suggestions for Authors

Thank you for inviting me to evaluate the article " Leucocytes telomere length and lung cancer risk: a systematic review and meta-analysis of prospective studies." This paper investigated the relationship between TL in leukocytes (LTL) and lung cancer risk. To compile the data and arrive at a more precise estimation of LTL's impact on lung cancer incidence, they conducted this systematic review and meta-analysis of prospective trials. The findings of this study can be used to identify individuals who are at a high risk of developing lung cancer and to implement preventive measures, especially for participants who smoke.

The Introduction part is too brief on the reasons for selecting prospective cohort studies and nested case-control studies, as well as the factors impacting TL measurements, so please expand on it.

The paper is well-written, with more comprehensive data and logical reasoning. So I suggest this paper be accepted after minor revisions.

Author Response

We agree with the reviewer, therefore we have revised the manuscript adding further explanation in the introduction (highlighted in red) as to why we selected prospective cohort studies and nested case-control studies in our analysis. A further addition also concerns the mention of other factors that can have an impact on TL measurements.